# Supplementation with Vitamin D_3_ Protects against Mitochondrial Dysfunction and Loss of BDNF-Mediated Akt Activity in the Hippocampus during Long-Term Dexamethasone Treatment in Rats

**DOI:** 10.3390/ijms241813941

**Published:** 2023-09-11

**Authors:** Daria Korewo-Labelle, Mateusz Jakub Karnia, Dorota Myślińska, Jan Jacek Kaczor

**Affiliations:** Department of Animal and Human Physiology, Faculty of Biology, University of Gdansk, 80-308 Gdansk, Poland; daria.korewo@phdstud.ug.edu.pl (D.K.-L.); mateusz.karnia@ug.edu.pl (M.J.K.); dorota.myslinska@ug.edu.pl (D.M.)

**Keywords:** glucocorticosteroids, cholecalciferol, neurogenesis, neuroprotection, energy metabolism

## Abstract

Dexamethasone (DEXA) is a commonly used steroid drug with immunosuppressive and analgesic properties. Unfortunately, long-term exposure to DEXA severely impairs brain function. This study aimed to investigate the effects of vitamin D_3_ supplementation during chronic DEXA treatment on neurogenesis, mitochondrial energy metabolism, protein levels involved in the BDNF-mediated Akt activity, and specific receptors in the hippocampus. We found reduced serum concentrations of 25-hydroxyvitamin D_3_ (25(OH)D_3_), downregulated proBDNF and pAkt, dysregulated glucocorticosteroid and mineralocorticoid receptors, impaired mitochondrial biogenesis, and dysfunctional mitochondria energy metabolism in the DEXA-treated group. In contrast, supplementation with vitamin D_3_ restored the 25(OH)D_3_ concentration to a value close to that of the control group. There was an elevation in neurotrophic factor protein level, along with augmented activity of pAkt and increased citrate synthase activity in the hippocampus after vitamin D_3_ administration in long-term DEXA-treated rats. Our findings demonstrate that vitamin D_3_ supplementation plays a protective role in the hippocampus and partially mitigates the deleterious effects of long-term DEXA administration. The association between serum 25(OH)D_3_ concentration and BDNF level in the hippocampus indicates the importance of applying vitamin D_3_ supplementation to prevent and treat pathological conditions.

## 1. Introduction

Dexamethasone (DEXA) is a synthetic glucocorticosteroid (GC) with potent and long-lasting immunosuppressive and analgesic effects. Due to these properties, DEXA is widely used in medicine for recovery after surgery, during chemotherapy, and in treating severe acute respiratory syndrome coronavirus 2 (SARS-CoV-2) and autoimmune diseases [1,2,3,4].

Under natural conditions, GCs are produced by activating the hypothalamic–pituitary–adrenal (HPA) axis and play essential roles in neuroendocrine regulation. In addition, GCs control many physiological processes, such as carbohydrate metabolism, immune maintenance, and cardiovascular activity [5]. Moreover, GCs are responsible for normal brain function and mitochondrial energy metabolism regulation [6].

Overactivity of the HPA axis results in excessive GC secretion, adrenal hypertrophy, and atrophy of certain organs such as the thymus, spleen, and skeletal muscles [7,8,9]. Moreover, long-term treatment or high-dose exposure to GCs also affects the neurons, causing a reduction in mitochondrial oxidation, mitochondrial membrane potential, and calcium retention capacity in cortical neurons [10]. Furthermore, there is sufficient evidence of an association between high GC levels, psychiatric disorders, and neurodegenerative diseases such as Alzheimer’s disease (AD) and Parkinson’s disease (PD) [11,12], which are disorders where mitochondria play a crucial role [13]. Like prolonged stress, long-term DEXA treatment carries many adverse side effects and causes significant damage. Studies show that long-term DEXA treatment also has a negative impact on the central nervous system (CNS), including the hippocampus [14,15,16]. The hippocampus, a limbic system structure, appears to be particularly sensitive to the effects of GCs. High levels of GCs cause morphological, metabolic, and functional changes [17,18,19]. It is indicated that DEXA treatment causes changes in the differentiation [20] and morphology of neurons in the hippocampus [21]. Therefore, it may also lead to depression, mood disorders, decreased concentration, insomnia, abnormal behavior, memory deficits, and altered nervous activity [15,16]. Chronic exposure to GCs may negatively affect the brain, impair hippocampal function, and reduce the volume of this brain region [22].

Moreover, some studies have also shown that excessive flow of exogenous or endogenous GCs can induce neuroplasticity in the hippocampus, mediated by a brain-derived neurotrophic factor (BDNF). Chronic stress has been found to reduce BDNF in the hippocampus. Epigenetic, transcriptomic, and proteomic regulation, mediated by BDNF and/or its receptor Tropomyosin receptor kinase B (TrkB) expression, has been suggested to underlie the observed changes in rodents and humans [23,24]. Therefore, a reduction in the BDNF/TrkB signaling pathway in the hippocampus may be associated with synaptic loss, decreased neuronal plasticity, and regression network connectivity [25,26]. On the other hand, insulin-like growth factor 1 (IGF-1) is a crucial mediator of brain development in both young and adult individuals [27]. It is suggested that circulating IGF-1 triggers negative feedback amplification in the HPA axis. According to Baldini et al., an increase in IGF-1 levels may be correlated with an increase in GR expression [28]. Although the effects of DEX on IGF-1 are not well understood, it has been suggested that GCs may regulate IGF-1 signaling pathways in the hippocampus [29].

The hippocampus is equipped with two types of steroid receptors: glucocorticoid (GR) and mineralocorticoid (MR) receptors [30,31]. Both receptors play a crucial role in maintaining the proper structure and function of the hippocampus. Any changes in their activity or ratio may increase the intensity of hippocampal damage [32]. Furthermore, chronic exposure to GCs may alter the expression of GR and MR or the MR/GR ratio, which may also impact mitochondrial function in the brain. Depending on the dose and/or duration of exposure, GCs may either increase or decrease the levels of MR and GR in the hippocampus, resulting in contrasting effects [32,33,34]. GR translocation to mitochondria directly affects their function. During long-term administration, GCs may lead to respiratory chain dysfunction, reduced ATP production, and abnormal mitochondrial biogenesis, ultimately increasing the cell sensitivity to death [34].

Vitamin D, known for its pleiotropic effects, regulates calcium and phosphate metabolism and cares for the skin and skeletal health, and it is also an immunomodulatory hormone [35]. The active form of vitamin D, 1,25-dihydroxy vitamin D_3_ (1,25(OH)_2_D_3_), binds to the vitamin D receptor (VDR) and genetically regulates various cellular processes by promoting heterodimerization of this retinoid-X receptor (RXR). The activity of the VDR-RXR complex affects the gene expression of many tissues, including skeletal muscles and the brain [36]. Interestingly, VDR is also found in mitochondria, where it plays a crucial role in maintaining their integrity and proper function [37]. This characteristic is fundamental for optimal brain function and development [38]. Studies have demonstrated that excessive GC levels may induce a depression-like state and reduce the protein levels of VDR and cytochrome P450 proteins involved in vitamin D activation and catabolism in brain regions such as the hippocampus and prefrontal cortex [39,40]. Remarkably, deficiencies in vitamin D are associated with decreased hippocampal volume. However, supplementation with vitamin D_3_ has been shown to prevent tissue atrophy and memory deficits under conditions of GC exposure [8,41,42], suggesting its potential involvement in a protective mechanism against the negative effects of stress on the brain [39].

Therefore, this study aimed to determine whether vitamin D_3_ supplementation would positively affect the BDNF-mediated Akt activity pathway and mitochondria energy metabolism in the rat hippocampus during long-term DEXA treatment.

## 2. Results

### 2.1. Vitamin D Concentration

A 28-day treatment with DEXA reduced serum 25(OH)D_3_ concentration in the DEX group (6.02 ± 0.86) compared with the CON group (10.97 ± 1.71; *p* < 0.01). After vitamin D_3_ supplementation, a significantly higher 25(OH)D_3_ concentration (12.99 ± 1.42 ng/mL) was found in the DEX + VD group compared with the DEX group (*p* < 0.001; Figure 1).

### 2.2. Body and Hippocampus Mass Changes

After long-term treatment of DEXA, body weight was significantly lower in both groups treated with DEXA (*p* < 0.0001). The body weight of rats in the DEX group (245.83 ± 8.07 g) was significantly lower by 39% compared with the CON group (405.20 ± 18.98 g). In the DEX + VD group, body weight significantly increased by 11% (277.57 ± 7.49 g) compared with the DEX group (*p* < 0.05; Figure 2a). The weight of the hippocampus in the DEX group (0.06 ± 0.03 g; *p* < 0.05) decreased significantly compared with the CON group (0.14 ± 0.03). However, after vitamin D_3_ supplementation in the DEX + VD group, the weight of the hippocampus (0.12 ± 0.02 g) significantly increased compared with the DEX group (Figure 2b; *p* < 0.05). We also observed changes in the hippocampus/body weight ratio, where the ratio was significantly higher in the DEX + VD group (0.044 ± 0.006) compared with the DEX group after vitamin D_3_ supplementation. However, we did not observe any differences in the hippocampus/body weight ratio in the DEX group (0.027 ± 0.006) compared to the CON group (0.034 ± 0.007) (Figure 2c).

### 2.3. Hippocampal GR, MR, and VDR Levels

After 28 days of DEXA treatment, no significant differences in VDR and MR protein levels were observed (Figure 3a,b). We found a statistically significant reduction in the GR protein level by 50% in the DEX group compared with the CON (*p* < 0.05), but no changes were found in the DEX + VD group or between groups. However, a trend suggesting an increase in GR level was observed in the DEX + VD group versus the DEX group (*p* = 0.063; Figure 3c).

### 2.4. Assessment of BDNF-Akt Signaling Molecules

To determine the changes in prosurvival proteins in the rat hippocampus during long-term administration of DEXA, we measured the Akt protein level and phosphorylated Akt (pAkt). There were no significant changes in Akt protein levels between the groups (Figure 4a). However, the phosphorylation state of Akt (p-Akt) significantly increased in the DEX + VD group compared with the DEX group. Moreover, the p-Akt level was almost two-fold higher in the DEX + VD group compared with the CON group, but this was not statistically significant (Figure 4b). The pAkt/Akt ratio showed a significant increase in the DEX + VD group compared with the DEX group. Still, there was no significant difference between the treatment groups and the CON group (Figure 4c).

To evaluate neurogenesis, we measured the protein levels of the precursor of brain-derived neurotrophic factor (proBDNF) and mature BDNF (mBDNF). After 28 days of DEX treatment, the level of proBDNF in the DEX group significantly decreased compared with the CON group. In contrast, we did not observe any statistically significant changes in the DEX + VD group (Figure 4d). Furthermore, we observed significantly elevated mBDNF protein levels during prolonged exposure to DEX + VD in the hippocampal tissue compared with the DEX group (*p* < 0.05). This increase in mBDNF level was accompanied by an elevated phosphorylation state of Akt after vitamin D_3_ supplementation in the rat hippocampus during long-term DEXA treatment. However, there was no significant difference in values between the DEX and CON groups (Figure 4e). We did not observe any changes in the IGF-1 protein level between the groups (Figure 4f).

### 2.5. Assessment of Mitochondrial Biogenesis and Oxidative Metabolism

After 28 days of DEXA treatment, we found a higher level of cytochrome c oxidase subunit II coded by mitochondria (COX II) in the hippocampus in the DEX + VD group (1.019 ± 0.325), which increased by 37% compared with the DEX group (*p* < 0.05; Figure 5a). This may partially confirm earlier studies showing the influence of vitamin D in the modulation of mitochondrial metabolism [43,44,45]. There were no significant changes between groups compared to the CON group (0.732 ± 0.069 A.U.). The nuclear subunit IV of cytochrome c oxidase (COX IV) protein levels was not significantly different between groups (Figure 5b). Furthermore, long-term treatment with DEXA resulted in a decrease in the peroxisome proliferator-activated receptor gamma coactivator 1-alpha (PGC-1α) protein level in the DEX rats when compared with the CON group. At the same time, supplementation with 600 IU of vitamin D_3_ caused no significant changes in protein level versus the DEX + VD group (Figure 5c).

### 2.6. Mitochondrial Oxidative Metabolism Activity

After 28 days of daily DEXA treatment, the activity of citrate synthase (CS) was reduced in the hippocampus by 48% in the DEX group (1.37 ± 0.44) when compared with the CON group (2.50 ± 0.33 µmol/min/mg of protein; *p* < 0.05). Moreover, CS activity was also significantly elevated by 42% in the DEX + VD group (2.35 ± 0.73 µmol/min/mg of protein) compared with the DEX group (*p* < 0.05; Figure 6a). However, we noticed no statistically significant changes in cytochrome c oxidase (COX) activity in the rat hippocampus despite a downward trend in the DEX group (57.53 ± 29.14) compared to CON (93.32 ± 26.07) and the upward trend in COX activity in the DEX + VD group (81.46 ± 32.48 nmol/min/mg of protein; Figure 6b).

## 3. Discussion

In the present study, we showed that long-term treatment with DEXA reduced the body mass of rats, induced hippocampus atrophy, and decreased proBDNF and GR protein levels in the rat hippocampus. We also found attenuated serum concentrations of 25(OH)D_3_ in DEXA-treated rats. Additionally, we observed a reduction in BDNF signaling in the hippocampus, a corticolimbic area in the brain. This reduction in signaling was associated with inhibited pAkt activity, decreased PGC-1α protein level, and disrupted mitochondrial energy metabolism. According to our results, systematic vitamin D_3_ supplementation partially reversed the adverse effects of DEXA. Thus, supplementation with vitamin D_3_ led to increased body and hippocampus mass, as well as an increased hippocampus/body mass ratio in rats treated with prolonged DEXA. Furthermore, the present study showed a restoration of 25(OH)D_3_ concentration to a value close to the control group. There was an elevation in neurotrophic factor protein level, along with augmented activity of pAkt, increased subunit II (coded by mitochondria protein level, and higher CS activity in the hippocampus of rats chronically treated with DEXA. We did not observe any changes in the protein levels of IGF-1 between the groups. This may suggest that the BDNF-mediated activity Akt is more susceptible to DEXA’s negative action than the IGF signaling pathway. Our findings demonstrate that supplementation with vitamin D_3_ plays a protective role in the corticolimbic area and partially mitigates the deleterious effects caused by long-term DEXA administration.

Our findings indicate differences in the protective effects of vitamin D between the hippocampus and body weight, the full understanding of which requires further study. One of the side effects of GC therapy is weight loss, which may manifest as osteoporosis, muscle wasting, and changes in fat composition [46,47]. The variations in the effectiveness of vitamin D3 supplementation in terms of its protective role in the hippocampus versus the rest of the body seem to arise from the multifaceted impacts of DEXA on metabolism. These impacts primarily focus on reducing fat mass, inducing muscle wasting, causing intramuscular fat deposition, and leading to mitochondrial dysfunction [48,49]. The mechanism(s) of DEXA’s action resemble stress responses and are associated with hormone dysregulation, insulin resistance, hepatic hexokinase inhibition, and inhibition of glucose oxidation. The consequence of subsequent exposure to GCs leads to increased lipolysis and the release of free fatty acids (FFAs) from adipose tissue and elevated proteolysis of skeletal muscle (decrease in PI3K markers, PKB/Akt, GSK3) dependent on the activation/inhibition of AMP-activated protein kinase (AMPK) by GCs. Reported evidence after 6 weeks of vitamin D_3_ supplementation in patients with hypercortisolemia had an effect on lipid profile and insulin sensitivity but did not significantly affect baseline body weight (BMI, waist circumference) [50].

Recently, it was reported that long-term treatment with DEXA has a negative impact on the CNS [51]. Similar alterations were also observed during a chronic stress response [52]. These changes may be associated with an appropriate balance of GR and MR levels and activity, which are crucial for the proper function of the CNS. Notably, data on GR and MR levels after DEXA treatment are inconsistent. On the one hand, studies showed that long-term administration of GCs reduced GR expression, increased neuroinflammation, and caused neurodegeneration in the hippocampus [53,54]. Another study showed changes in mRNA expression in both receptors, revealing that GR was decreased while MR was increased in the hippocampus, striatum, and prefrontal cortex in DEXA-treated mice [55]. In line with this finding, our data showed similar trends in the hippocampus. On the other hand, the data reported inconsistent results regarding changes in GR and MR protein levels and gene expression in response to GCs action [10,34,56,57,58]. The results published by Zhe and coworkers showed that single-prolonged stress causes a reduced expression of MR and GR in CA1 of the hippocampus [32]. The differences between our results and those of other studies may be attributed to the dose and duration of GC exposure or other factors that can have different effects [10,59]. Short-term application of both high and low doses of GCs similarly affects GR in cortical neurons by increasing its mitochondrial localization, while during long-term supply, only a high dose decreases GR levels in mitochondria and negatively affects mitochondrial function in neurons [10]. In the current study, we found a lower GR protein level after long-term DEX treatment with the protective effect of vitamin D_3_ supplementation. Although supplementation with vitamin D_3_ partially reversed this adverse impact, the effect was not statistically significant.

In this study, we showed that the 28-day administration of DEXA at a dose of 2 mg/kg/day caused a decrease in the serum concentration of 25(OH)D_3_. A previous study found that DEXA affects VDR in a GR-dependent manner by increasing *Vdr* transcripts de novo [60]. Surprisingly, we did not find any substantial differences in VDR protein levels with DEX treatment alone or combined with D_3_ supplemented when compared to the control group. Our finding is partially in line with the report by Jiang et al. [39], which demonstrated dysregulation in vitamin D metabolism caused by a 10-day DEXA supply. The authors observed a decrease in CYP27B1, CYP24A1, and VDR expression not only in the hippocampus but also in the prefrontal cortex, heart, and kidneys. Moreover, VDR activity is essential at the cellular level for cell differentiation, cell growth, and apoptosis [61]. However, VDR silencing induces unbalanced metabolism and ultimately contributes to cytotoxicity, while VDR ablation induces many negative consequences, including mitochondrial damage and premature cell death [37]. Additionally, VDR may have an impact on the mitochondrial respiratory chain [62] by transcribing the mitochondrial (COX II) and nuclear (COX IV) subunits of cytochrome c oxidase, which are involved in ATP synthesis [37]. Our present study showed that DEXA significantly lowered the COX subunit II protein level, while vitamin D_3_ supplementation effectively reversed this alteration. Mitochondrial subunits play a crucial role in the catalytic cycle of the enzyme, with nuclear subunits being associated with regulating enzyme activity and ensuring the structural stability of the complex [63]. In our study, we observed a reduction in COX subunit II but not COX subunit IV. In addition, we did not find any changes in COX activity. Desquiret et al. [64] reported that chronic DEXA administration leads to a decrease in the activity of the electron transport chain (ETC) complex I and II while also increasing the activity of complex IV. To determine the function of mitochondria in the Krebs cycle, we examined the activity of citrate synthase (CS). CS is an enzyme localized in the mitochondrial matrix and is used as a quantitative enzymatic marker for the presence of intact mitochondria [65,66]. Our data indicate that long-term DEXA administration decreased CS activity in the hippocampus, while vitamin D_3_ supplementation contributed to maintaining proper enzyme activity.

According to our results, long-term treatment with DEXA reduces the activity of mitochondrial energy metabolism, while supplementation with vitamin D_3_ restores its proper function in the hippocampus. Vitamin D shows a potential protective role and causes reverse oxidative stress-induced cognitive impairment, which was also presented by Hajiluian and coworkers [67]. This translates into our study results, wherein, in the vitamin D_3_-supplemented group, there was a noticeable improvement in the function of the ETC and the citric acid cycle in the hippocampus. Vitamin D_3_ supplementation or administration of its metabolites reduces neurological damage and neurotoxicity. The detailed biochemical mechanism is still unclear, but it is known that vitamin D can act on multiple pathways, such as neuronal antioxidant pathways, immunomodulation, calcium regulation, and glutamatergic systems [68]. All these data demonstrate an antagonistic effect of vitamin D on GC activity [69]. Given that the impairment of mitochondrial function plays an important role in response to stress, brain aging, and the early stages of neurodegenerative diseases, we believe that enhancing mitochondrial brain energy metabolism with vitamin D may be an essential protective mechanism [66,70]. This study found a positive influence of vitamin D_3_ supplementation on the BDNF protein level, i.e., a higher serum concentration of 25(OH)D_3_ upregulated the BDNF level in the hippocampus of prolonged DEXA-treated rats. The association between serum concentration of 25(OH)D_3_ and BDNF level in the hippocampus reported in this study may indicate the importance of improving vitamin D with supplementation, which acts as a strategy for preventing and treating cognitive decline and increasing BDNF in excessive-prolonged flow of exogenous or endogenous GCs (chronic stress). On the one hand, Yosefian et al. reported that vitamin D_3_ supplementation did not affect BDNF concentration in the hippocampus in animal models of depression [71]. On the other hand, another study showed that the modulation of hippocampal BDNF with vitamin D might be an effective prevention strategy against depression in animal models of depression [72].

We are very far from any speculation, but our results strongly suggest that vitamin D_3_ supplementation may reverse the negative changes induced by long-term DEXA treatment. This reversal leads to the upregulation of BDNF and nearly restores GR protein levels. Based on previously reported results [73] and our findings, we assume the existence of crosstalk between GR and BDNF in the hippocampus. Therefore, the present study showed that the modulatory effects of vitamin D during long-term exposure to DEXA on GR restored mitochondrial function, and BDNF-mediated Akt activity reversed neuropathophysiological changes in the hippocampus. Our findings may have significant implications for neuroplastic processes, cognitive function, and several neuropsychiatric disorders. Moreover, our results provide further evidence of a key link between vitamin D and BDNF-mediated Akt activity, as well as the restoration of physiological function in the mitochondria through GR in the hippocampus. These findings may contribute to the neuroprotective and therapeutic potential (abilities) of vitamin D in patients with neuropathophysiological disorders.

This study has some limitations. First, we did not estimate TrkB, IGR, or other proteins involved in the signaling pathway, which could have directly confirmed our findings in the hippocampus of DEXA-treated rats. Second, we did not measure protein levels that can form complexes with GR, such as HSP70/90, Bcl-2, or proteins targeting mitochondria associated with chaperones that aid in their mitochondrial translocation. This would have helped to demonstrate the meaningful role of GR in the modification of mitochondria-coded protein expression. These limitations were primarily due to the lack of sufficient tissue for the analysis.

Overall, our findings showed that supplementation with vitamin D_3_ partially reversed negative changes in the hippocampus induced with long-term DEXA treatment in rats. We observed upregulation of BDNF and nearly restored GR protein level after vitamin D_3_ administration. Consequently, the present study demonstrated a higher activity of pAkt, elevated COX subunit II protein level, and augmented activity of CS in the hippocampus, suggesting the modulatory abilities of vitamin D during long-term exposure to DEXA. These findings have significant implications for neuroplastic processes, cognitive function, and neuropathophysiological conditions. Moreover, these findings may contribute to the understanding of the neuroprotective and therapeutic abilities of vitamin D in patients with neurophysiological disorders.

## 4. Materials and Methods

### 4.1. Animals

The Local Ethics Committee in Bydgoszcz, Poland (No. 10/2019), approved our animal studies. All procedures were carried out following European guidelines.

Male Wistar rats (56–70 days postnatal) weighing 300–400 g were used in this study. The animals were obtained from the Academic Experimental Animal House at the Medical University of Gdansk, Poland. The rats were housed 3–4 per cage in climate-controlled conditions (temperature: 22 ± 2 °C; humidity: 55 ± 2%) with a 12:12 h light/dark cycle and provided food and water ad libitum.

### 4.2. Experimental Procedure

The rats were habituated to the various experimental procedures daily for two weeks before the experiment began. The eighteen animals were randomly assigned to three groups: control (CON; *n* = 5), DEXA placebo (DEX; *n* = 6), and DEXA supplemented with vitamin D_3_ (DEX + VD; *n* = 7). The experiment lasted for 28 days. Throughout this time, the CON group was treated intraperitoneally with saline, and the two other groups were treated with dexamethasone at 2 mg/kg/day (dexamethasone D4902, DEXA, Sigma-Aldrich, St Paul, MN, USA) in the same manner. The DEX + VD group was orally supplemented with vitamin D_3_ (Juvit D3, PPF HASCO-LEK. SA., Wrocław, Poland) at 600 IU/kg/day, and the DEX group was orally supplemented with vegetable oil as a placebo. Blood was collected on Day 1 and Day 28 of the experiment from the tail vein under isoflurane anesthesia. On Day 29 of the experiment, all animals were sacrificed using decapitation. The brains were collected, and the hippocampus was dissected, weighed, frozen in liquid nitrogen, and stored at −80 °C for further analysis.

### 4.3. Tissue Preparation

#### 4.3.1. Blood Collection

To determine the serum 25(OH)D_3_ concentration, blood samples were centrifuged for 10 min at 2000× *g* at 4 °C. The serum was collected and frozen at −80 °C for further analysis.

#### 4.3.2. Western Blot

A part of the hippocampus was homogenized in RIPA buffer (89901; Thermo Scientific, Waltham, MA, USA) in the presence of protease and phosphatase inhibitors (1:100; 1861280; Thermo Scientific, Waltham, MA, USA). The 8% tissue homogenates were centrifuged at 750× *g* for 10 min at 4 °C. Then, the collected supernatant was centrifuged again at 12,000× *g* (10 min at 4 °C), aliquoted to cryogenic microfuge tubes, and stored at −80 °C for further analysis.

#### 4.3.3. Enzymes Activity

The rest of the hippocampi were homogenized in buffer containing 50 mM Tris-HCl (T3253; Sigma-Aldrich, USA), 150 mM NaCl, 1 mM EDTA (ED3SS; Sigma-Aldrich, USA), 0.5 mM DTT (443853B; VWR International; Radnor, PA, USA) pH 7.2, and 0.2% HALT protease inhibitors cocktail (P834; Sigma Aldrich, STL, USA). The 4% tissue homogenates were centrifuged at 750× *g* for 10 min at 4 °C. The supernatant was frozen and stored at −80 °C for further analysis.

### 4.4. Vitamin D_3_ Metabolite Concentration

The concentration of 25(OH)D_3_ in serum was measured using liquid chromatography coupled to tandem mass spectrometry (LC-MS/MS), according to the procedure of Rola and coworkers [74]. Briefly, serum was analyzed using the Eksigent Exion LC HPLC system with a CTC PAL autosampler (Zwinger, Hofstetten bei Brienz, Switzerland) coupled to a QTRAP^®^ 4500 MS/MS system (Sciex, Framingham, MA, USA).

### 4.5. Protein Expression

Western Blot Analysis

Samples (approximately 25–30 µg) were prepared from the collected supernatants and mixed with RIPA buffer. The protein samples were denatured with heating at 95 °C for 5 min, cooled to room temperature, and then separated into 10% and 12% mini protean TGX precast Protein Gels (4561035, 4561045; BioRad, Hercules, CA, USA). Membranes of thickness 0.2 μm were used for semi-dry transfer. The membranes were blocked with Every Blot Blocking Buffer (12010020; BioRad, CA, USA) or 5% nonfat milk in TBST (for the determination of PGC-1*α*). The primary antibodies used were: VDR (ab3508; diluted 1:1000, Abcam, Cambridge, UK), MR (ab64457; diluted 1:1000, Abcam, UK), GR (ab183127; diluted 1:500, Abcam, UK), BDNF (ab108319; diluted 1:1000, Abcam, UK), Akt (C67E7; diluted 1:1000, Cell Signaling, Danvers, MA, USA), pAkt (D25E6; diluted 1:1000, Cell Signaling, MA, USA), IGF-1 (ab9572; diluted 1:1000, Abcam, UK), COX II (NBP2-94364; diluted 1:1000, Novusbio, Centennial, CO, USA), COX IV (4D11-B3-E8; diluted 1:1000, Cell Signaling, MA, USA), PGC-1α (ab191838; diluted 1:500, Abcam, UK), and ß-tubulin. (AF7011; diluted 1:1000, Affinity, Beachwood, OH, USA). The membranes were treated with secondary rabbit anti-mouse (ab6728; Abcam) and goat anti-rabbit (111-035-003; Jackson ImmunoResearch, Ely, UK) antibodies (both diluted 1:1000–1:3000). The antibodies were prepared according to the instructions provided by the manufacturer. The results were visualized using Clarity Western ECL Substrate (1705061; BioRad, CA, USA) and imaged using the ChemiDoc MP imaging system (BioRad, CA, USA). Protein concentration was measured with the Pierce™ BCA protein assay method.

### 4.6. Enzyme Activities

#### 4.6.1. Citrate synthase Activity

Citrate synthase (CS) activity was determined according to Dzik and coworkers [43]. In brief, 3 μL of the supernatant was combined with 181 μL of buffer (50 mM TRIS-HCl with 5 mM EDTA, pH 8.1), 20 μL of freshly made DTNB (1 mM), 2 μL of acetyl-coenzyme A (15 mM), and 2 μL of freshly made oxaloacetate acid (10 mM) to initiate the reaction. The results were obtained in 2 min by measuring the absorbance change at 412 nm using a multimode microplate reader (Varioskan Flash-Spectral Scanning Multimode Microplate Reader 183, Thermo Fisher Scientific, MA, USA) at 37 °C. The CS activity was measured in duplicate and expressed as µmol/min/mg of protein.

#### 4.6.2. Cytochrome c Oxidase Activity

Cytochrome c oxidase (COX) activity was measured according to [75] using a microplate combining 5 μL of supernatant, 192 μL of K phosphate buffer (50 mM, 1 mM EDTA, pH 7.2), and 3 μL of 2 mM reduced cytochrome c (c2037; Sigma-Aldrich; USA; reduced with ascorbic acid 20:1) to initiate the reaction. The COX activity was measured in duplicate at 37 °C, with an absorbance of 550 nm using a microplate reader (Varioskan Flash-Spectral Scanning Multimode Microplate Reader 183, Thermo Scientific, MA, USA). The results were expressed as nmol/min/mg of protein. Protein concentration was measured with the Bradford method.

### 4.7. Statistical Analysis

All results were analyzed using the GraphPad Prism 8.3 software program (GraphPad Software, San Diego, CA, USA). A one-way ANOVA with least significant difference (LSD) post hoc test was used to analyze body and hippocampus weight, 25(OH)D_3_ concentration, and the results obtained from the Western Blot with chemiluminescence using Image Lab 6.1.0 software. A one-way ANOVA with a Tukey post hoc test was used for CS and COX enzyme activity. *p*-values < 0.05 were considered significant. Data are presented as mean ± SEM.

## Figures and Tables

**Figure 1 ijms-24-13941-f001:**
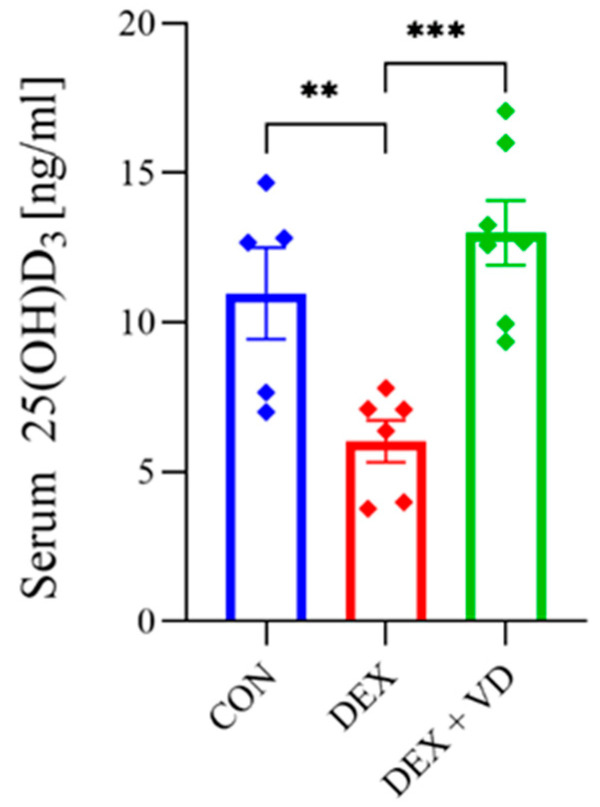
The concentration of 25(OH)D_3_ in the serum in the control group (CON, *n* = 5), the DEXA-treated group (DEX, *n* = 6), and the group supplemented with vitamin D_3_ (DEX + VD, *n* = 7) after 28 days of the experiment. Results are expressed as mean ± SEM. ** *p* < 0.01; *** *p* < 0.001.

**Figure 2 ijms-24-13941-f002:**
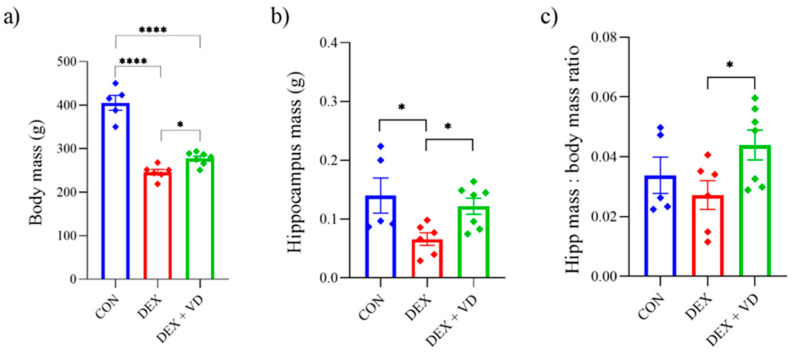
The body weight (**a**), hippocampus mass (**b**), and the hippocampus to body mass ratio (**c**) of rats in the CON (*n* = 5), DEX (*n* = 6), and DEX + VD (*n* = 7) groups. Results are expressed as mean ± SEM. * *p* < 0.05; **** *p* < 0.0001.

**Figure 3 ijms-24-13941-f003:**
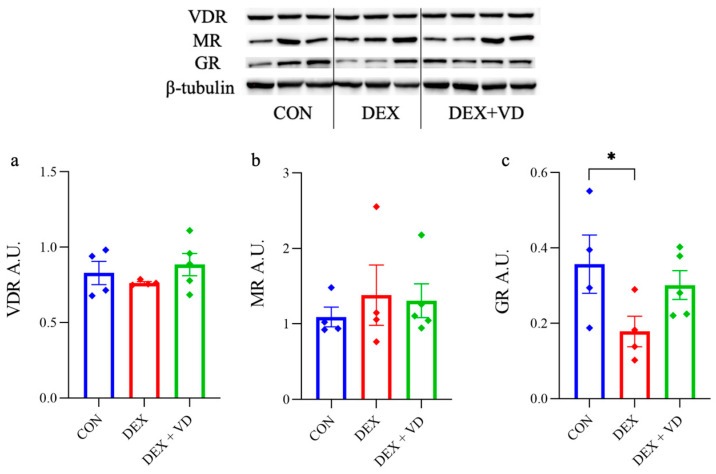
The protein levels of VDR (**a**), MR (**b**), and GR (**c**) in the hippocampus of rats in the control group (CON, *n* = 4), the DEXA-treated group (DEX, *n* = 4), and the group supplemented with vitamin D_3_ (DEX + VD, *n* = 5). Results are expressed as the mean ± SEM; * *p* < 0.05.

**Figure 4 ijms-24-13941-f004:**
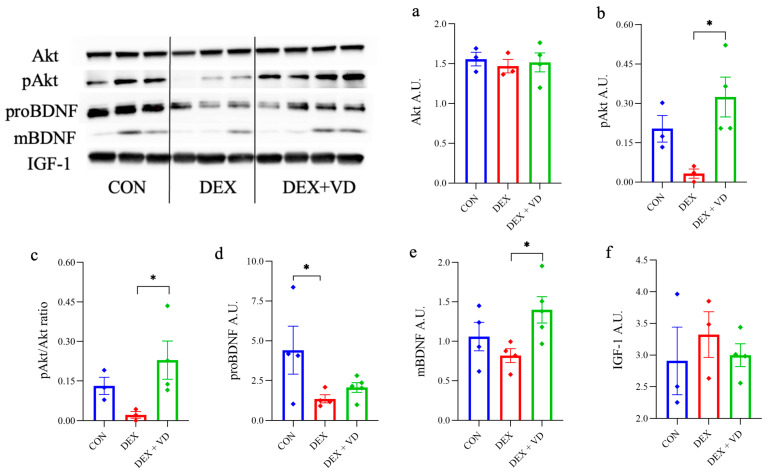
The protein levels of Akt (**a**), pAkt (**b**), proBDNF (**d**), mBDNF (**e**), and IGF-1 (**f**) and the pAkt/Akt ratio (**c**) in rat hippocampus in the control group (CON) and groups treated with DEXA and supplemented with vitamin D_3_ (DEX + VD). Results are expressed as the mean ± SEM. * *p* < 0.05.

**Figure 5 ijms-24-13941-f005:**
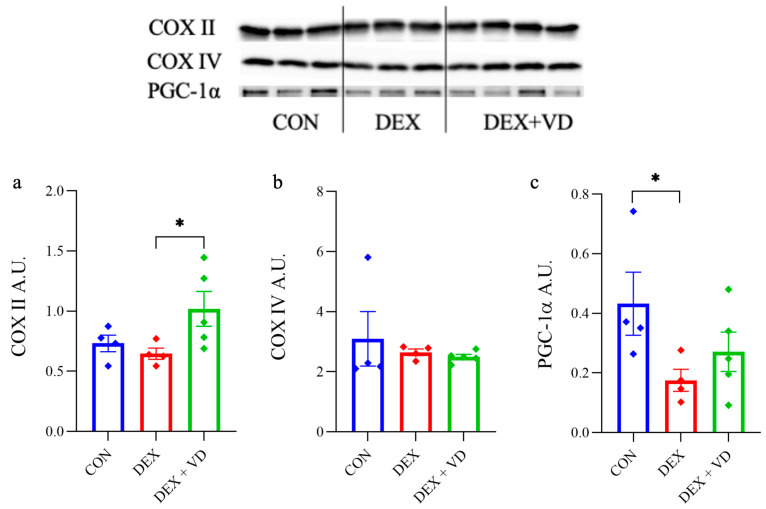
The protein levels of COX subunits (mitochondrial-II and nuclear-IV) (**a**,**b**) and peroxisome proliferator-activated receptor gamma coactivator 1-alpha (PGC-1α) (**c**) in rats from the control group (CON, *n* = 4), the DEXA-treated group (DEX, *n* = 4), and the group supplemented with vitamin D_3_ (DEX + VD, *n* = 4). Results are expressed as the mean ± SEM. * *p* < 0.05.

**Figure 6 ijms-24-13941-f006:**
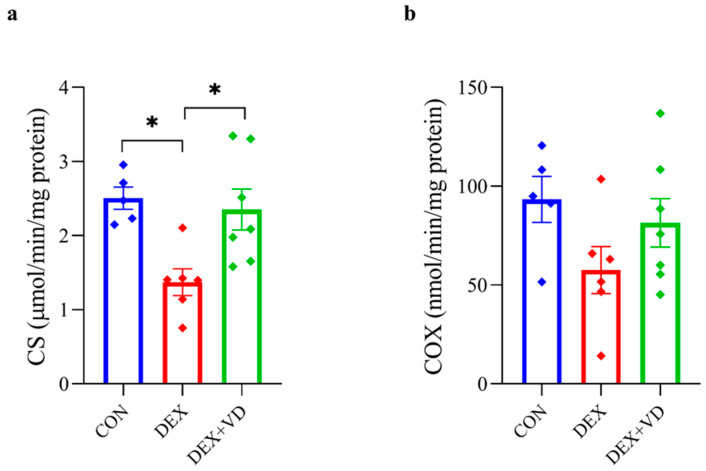
The CS (**a**) and COX (**b**) activity in the hippocampus of rats from the control group (CON, *n* = 5), the DEXA-treated group (DEX, *n* = 6), and the group supplemented with vitamin D_3_ (DEX + VD, *n* = 7). Results are expressed as the mean ± SEM. * *p* < 0.05.

## Data Availability

The datasets used and/or analyzed in the current study are available from the corresponding author upon reasonable request.

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
