# Peer review of "Supplementation with Vitamin D3 Protects against Mitochondrial Dysfunction and Loss of BDNF-Mediated Akt Activity in the Hippocampus during Long-Term Dexamethasone Treatment in Rats"

_ijms, 2023, doi:10.3390/ijms241813941_

Round 1

Reviewer 1 Report

The absence of a vitamin D3 control group in your experiment introduces several potential drawbacks and limitations. Firstly, the lack of a group receiving only vitamin D3 supplementation, distinct from dexamethasone, undermines the ability to confidently attribute observed effects to the combination of dexamethasone + vitamin D3 versus vitamin D3 alone. Even the normal group fails to account for potential standalone effects of vitamin D3. Furthermore, the experiment's inability to discern potential interactions, additive or synergistic effects between dexamethasone and vitamin D3 when administered in isolation versus together, hinders a comprehensive understanding. Precisely quantifying the magnitude of vitamin D3's protective or mitigating effects on dexamethasone is compromised without a direct comparison to a vitamin D3 only group. Moreover, the mechanism of action of vitamin D3 remains partially obscured—whether it acts directly on dexamethasone-affected pathways or through independent mechanisms—highlighting the insights a vitamin D3 control group could offer. The study's generalizability to scenarios involving sole administration of vitamin D3 without confounding factors is limited, confining its applicability to the specific combination employed. Lastly, the risk of attributing confounding effects solely to vitamin D3 while overlooking other contributing factors is heightened, accentuating the necessity to isolate vitamin D3's independent contribution through appropriate control measures. It's essential to acknowledge that the combined effects of dexamethasone and vitamin D3 exhibited more pronounced impacts on certain measured parameters compared to the normal group. Consequently, it becomes imperative to specifically isolate and discern the effects of vitamin D3 on the group of normal rats.

very good 

Author Response

Dear Reviewer,

The authors of the manuscript would like to thank you for all your comments and suggestions concerning improvements of this paper. We followed them to revise the manuscript or tried to explain any ambiguities in the point-by-point response.

Comment 1. The absence of a vitamin D3 control group in your experiment introduces several potential drawbacks and limitations. Firstly, the lack of a group receiving only vitamin D3 supplementation, distinct from dexamethasone, undermines the ability to confidently attribute observed effects to the combination of dexamethasone + vitamin D3 versus vitamin D3 alone. Even the normal group fails to account for potential standalone effects of vitamin D3. Furthermore, the experiment's inability to discern potential interactions, additive or synergistic effects between dexamethasone and vitamin D3 when administered in isolation versus together, hinders a comprehensive understanding. Precisely quantifying the magnitude of vitamin D3's protective or mitigating effects on dexamethasone is compromised without a direct comparison to a vitamin D3 only group. Moreover, the mechanism of action of vitamin D3 remains partially obscured—whether it acts directly on dexamethasone-affected pathways or through independent mechanisms—highlighting the insights a vitamin D3 control group could offer. The study's generalizability to scenarios involving sole administration of vitamin D3 without confounding factors is limited, confining its applicability to the specific combination employed. Lastly, the risk of attributing confounding effects solely to vitamin D3 while overlooking other contributing factors is heightened, accentuating the necessity to isolate vitamin D3's independent contribution through appropriate control measures. It's essential to acknowledge that the combined effects of dexamethasone and vitamin D3 exhibited more pronounced impacts on certain measured parameters compared to the normal group. Consequently, it becomes imperative to specifically isolate and discern the effects of vitamin D3 on the group of normal rats.

Response 1. Thank you for such an extensive comment related to this topic. Indeed, the vitamin D3-supplemented control group was not included in the study conducted. However, in our parallel study, where we have not yet published the results, it was considered. In those studies, we did not observe statistically significant differences in hippocampal mass between the control group (CON) and the vitamin D3-supplemented control group (VD). Moreover, we did not observe synergistic effects between vitamin D3 and dexamethasone. The following are the results of yet-unpublished studies. Twenty-six animals participated in the experiment; however, some of the material was used for immunohistochemical measurements (brains were perfused and determination of tissue mass was impossible).

CON: n=3; VD: n=3; DEX: n=4; DEX+VD: n=4

In addition, dexamethasone is considered a synthetic glucocorticosteroid, and the literature clearly indicates its antagonistic effect on vitamin D. What is shown i.e. in the study of Obradovic et al. where a significant difference in survival, apoptosis, neurite number, and MAPK pathway activation of cells in the primary culture of the rat hippocampus was observed. In addition, the antagonizing effect of vitamin D on certain glucocorticoid-mediated actions was also apparent. [Obradovic et al. 2006].

Obradovic D, Gronemeyer H, Lutz B, Rein T. Cross-talk of vitamin D and glucocorticoids in hippocampal cells. J Neurochem. 2006 Jan;96(2):500-9. doi: 10.1111/j.1471-4159.2005.03579.x. Epub 2005 Nov 29. PMID: 16336217.

Sincerely,

Jan J Kaczor

Reviewer 2 Report

This manuscript presents findings on the potential of vitamin D3 supplementation to mitigate the harmful impacts of prolonged dexamethasone treatment on the hippocampus of rats. The study is intriguing, and the authors have effectively elucidated their research. Nevertheless, some minor revisions are requisite prior to its publication, and the following are my remarks:

1.      Throughout the manuscript, there are several language mistakes. Therefore, I recommend a professional round of language editing before the paper is published.

2.      Line no. 75-76 – “Any changes in their activity or ratio may increase the sensibility of hippocampal damage.” This sentence can be corrected to “Any changes in their activity or ratio may increase the intensity of hippocampal damage.”

3.      For western blot results “Protein content” has been used in the manuscript. Please use either “protein expression” or “protein levels.” Western blot technique is not used to measure the protein content.

4.      Line no. 174 - instead of “during 28-days of Dexa treatment”, please use “after 28-days of Dexa treatment.”

5.      Introduction is not justified.

6.      Font size is different in some paragraphs.

7.      Line no. 223 - “recently it was indicated” can be corrected to “recently it was reported.”

8.      Please rewrite line no. 232, as “in line with this finding, our data showed similar trends in hippocampus.”

9.      Line no. 254-255 – “Moreover, the VDR activity is essential at the cellular level for cell differentiation, cell growth, and others”, sentence is incomplete.

10.  Material and method section requires extensive rewriting:

a.    Age of the rats (postnatal day) has not mentioned.

b.    It is not clear which is the last day of the investigation.

c.     Please rewrite line no. 337

d.    Line no. 342 - “animals were habituated” is incorrect because there is no behavior study done.

e.     Line no. 345 - “treated with saline by abdominal injection” instead write “treated intraperitoneally” is more appropriate.

f.      It is not clear if the Dexa injection was also given abdominally/ intraperitoneally.

g.    Line no. - 349 - “collected on 1 and 28 days” is incorrect. It should be “on Day1 and Day 28.”

h.    Line no. 351 - “Brains were detached, and hippocampus was removed” is incorrect. It should be “Brains were collected, and hippocampus was dissection.”

i.      Line no. 356 - instead of “aliquoted” “collected” would be appropriate.

j.      Line no. 361 - instead of “microtubes” cryogenic microfuge tubes would be correct.

This manuscript has several language and grammatical errors. Particularly, the "Materials and Methods" section contains numerous instances of such errors, as indicated in my comments. A professional round of language editing is strongly recommended.

Author Response

Dear Reviewer,

The authors of the manuscript would like to thank you for all your comments and suggestions concerning improvements of this paper. We followed them to revise the manuscript or tried to explain any ambiguities in the point-by-point response.

Comment 1. Throughout the manuscript, there are several language mistakes. Therefore, I recommend a professional round of language editing before the paper is published.

Response 1. We have carefully revised our manuscript. A professional English language editor has made the necessary changes.

Comment 2. Line no. 75-76 – “Any changes in their activity or ratio may increase the sensibility of hippocampal damage.” This sentence can be corrected to “Any changes in their activity or ratio may increase the intensity of hippocampal damage.”

Response 2. Thank you for your comment, we have changed this sentence as you suggested.

Line: 79-80: “Any changes in their activity or ratio may increase the intensity of hippocampal damage”

Comment 3. For western blot results “Protein content” has been used in the manuscript. Please use either “protein expression” or “protein levels.” Western blot technique is not used to measure the protein content.

Response 3. Thank you, "protein content" has been changed to "protein levels" in the manuscript. In addition, the descriptions of the graphs in the figures have been changed.

Comment 4. Line no. 174 - instead of “during 28-days of Dexa treatment”, please use “after 28-days of Dexa treatment.”

Response 4. Thank you for this, it has been changed.

Line 209-210: “After 28 days of DEXA treatment, we found a higher protein level”

Comment 5. Introduction is not justified.

Response 5. Thank you for the information, the introduction has been corrected.

Comment 6. Font size is different in some paragraphs.

Response 6. We apologize for the oversight. The font size has been adjusted in “Materials and Methods”.

Comment 7. Line no. 223 - “recently it was indicated” can be corrected to “recently it was reported.”

Response 7. I appreciate your comment. Part of this sentence has been changed according to your remarks.

Line 352: “Recently, it was reported that long-term treatment with DEXA has a negative impact on the CNS.”

Comment 8. Please rewrite line no. 232, as “in line with this finding, our data showed similar trends in hippocampus.”

Response 8. I greatly appreciate your comment. The sentence “Our data presented in the hippocampus are in line with these findings” has been changed to “In line with this finding, our data showed similar trends in the hippocampus.”

Comment 9. Line no. 254-255 – “Moreover, the VDR activity is essential at the cellular level for cell differentiation, cell growth, and others”, sentence is incomplete.

Response 9. Thank you for your insights. This sentence has been corrected as follows: Line Line 383-384: “Moreover, VDR activity is essential at the cellular level for processes such as cell differentiation, growth, and apoptosis”.

Comment 10. Material and method section requires extensive rewriting:

Comment 10a. Age of the rats (postnatal day) has not mentioned.

Response 10a. Thank you, we have added the age of the rats as you suggested.

Line 576: “Male Wistar rats (56-70 postnatal day) weighing 300-400 g were used in the study.”

Comment 10b. It is not clear which is the last day of the investigation.

Response 10b. Thank you for your comment. Indeed, the information was not clear. The experiment lasted 28 days, during which the animals were supplemented with vitamin D/placebo and treated with dexamethasone/saline. Thus, the animals were sacrificed a day after to ensure a full 28-day supplementation period. The sentence has been changed to:

Line 591-592: “On Day 29 of the experiment, all animals were sacrificed by decapitation.”

Comment 10c. Please rewrite line no. 337

Response 10c. The sentence has been rewritten according to your request:

Line 576-578: “Male Wistar rats (56-70 postnatal day) weighing 300-400 g were used in the study. The animals were obtained from the Academic Experimental Animal House of the Medical University of Gdansk, Poland”

Comment 10d. Line no. 342 - “animals were habituated” is incorrect because there is no behavior study done.

Response 10d. Thank you for the comment. Prior to the experiment, the animals were accustomed to the researchers and to procedures such as blood draws, vitamin D/placebo administration, and dexamethasone/saline injection. The goal of this process was to reduce the time taken to perform the procedures and to minimize the stress response in the animals during the procedures, which could affect the overall results.

Comment 10e. Line no. 345 - “treated with saline by abdominal injection” instead write “treated intraperitoneally” is more appropriate.

Response 10e. Thank you, we have corrected it according to your suggestion.

Line 585-588:The experiment lasted 28 days. Throughout this time, the CON group was injected intraperitoneally saline, and the other two groups were administered dexamethasone at a dose of 2 mg/kg/day (Dexamethasone D4902, DEXA, Sigma–Aldrich, MN, USA) in the same manner.”

Comment 10f. It is not clear if the Dexa injection was also given abdominally/ intraperitoneally.

Response 10f. Indeed, it was not clearly specified how dexamethasone was administered. This information was carried in the manuscript:

Line 585-588:The experiment lasted 28 days. Throughout this time, the CON group was injected intraperitoneally saline, and the other two groups were administered dexamethasone at a dose of 2 mg/kg/day (Dexamethasone D4902, DEXA, Sigma–Aldrich, MN, USA) in the same manner.”

Comment 10g. Line no. - 349 - “collected on 1 and 28 days” is incorrect. It should be “on Day1 and Day 28.”

Comment 10h. Line no. 351 - “Brains were detached, and hippocampus was removed” is incorrect. It should be “Brains were collected, and hippocampus was dissected.”

Response 10g I 10h. Thank you, we have corrected the form of the sentences according to your suggestion

Line 590: Blood was collected on the Day 1 and Day 28 of the experiment”

Line 592: “Brains were collected, and the hippocampus was dissected”

Comment 10i. Line no. 356 - instead of “aliquoted” “collected” would be appropriate.

Comment 10j. Line no. 361 - instead of “microtubes” cryogenic microfuge tubes would be correct.

Response 10i, 10j. Indeed, the vocabulary in the manuscript was not correct, so they were corrected as you suggested.

Line 597: “The serum was collected and frozen at -80°C for further analysis.”

Line 603-604: Then the collected supernatant was centrifuged again at 12 000 g (10 min at 4°C), aliquoted to cryogenic microfuge tubes, and stored at -80°C for further analysis.

The revised version of the manuscript with tracked changes enabled, allows you to see the specific revisions made in response to your comments. We hope that these revisions adequately address your concerns and suggestions.

Sincerely,

Jan J Kaczor

Reviewer 3 Report

The authors investigated the protective effect of vitamin D3 on the hippocampus during the long-term DEXA treatment. They concluded that vitamin D3 could reverse the negative impact on the hippocampus through long-term DEXA treatment by restoring mitochondria function and BDNF-mediated Akt activity. This manuscript is well-written, and the results are sound. However, a few points need to revise before it can be accepted for publication.

 1. Controls are missing in western blots for both Figure 4 and Figure 5.

 2. The difference in body mass for Con and DEX+VD is significant, while it is not apparent in hippocampus mass. This is a possible cause for the ratio of hippocampus mass/ body mass. This observation leads me to speculate that the effect of vitamin D3 is more effective in the hippocampus than in the whole body. What is the possible reason? The authors did not make any comment or discussion on this observation.

 3. In line 211, Freversed should be reversed. 

Author Response

Dear Reviewer,

The authors of the manuscript would like to thank you for all your comments and suggestions concerning improvements of this paper. We followed them to revise the manuscript or tried to explain any ambiguities in the point-by-point response.

Comment 1. Controls are missing in western blots for both Figure 4 and Figure 5.

Response 1. All determinations of selected proteins were made from a single sample. Each time, the protein content of hippocampal lysates was assessed before Western Blot analysis. The determinations were performed using the "Pierce BCA Protein Assay Kit" (23225, Thermo Scientific, USA) and were carried out according to the manufacturer's instructions. In addition, when performing the procedure for determining selected proteins (Western Blot with Chemiluminescence), one of the steps is the Stain Free reading, which allows quality assessment at each experiment stage. The Stain Free reading during our study showed an even protein content for the tested samples. Due to the lack of differences in protein readings before as well as during the execution of the experiment, the small size of the study material, and the normalizing of the results for total protein in the sample, we abandoned the additional determination of β-tubulin. One of the main reasons was the lack of tissue, therefore, we have made the decision to measure other important proteins involved in the described molecular pathways.

Comment 2. The difference in body mass for Con and DEX+VD is significant, while it is not apparent in hippocampus mass. This is a possible cause for the ratio of hippocampus mass/ body mass. This observation leads me to speculate that the effect of vitamin D3 is more effective in the hippocampus than in the whole body. What is the possible reason? The authors did not make any comment or discussion on this observation.

Response 2. Thank you for your comment regarding this topic. We apologize for this omission. We have added some sentences in the Discussion section as follows:

Our findings indicate differences in the protective effects of vitamin D between the hippocampus and body weight, the full understanding of which requires further study.  One of the side effects of GC therapy is weight loss, which may manifest as osteoporosis, muscle wasting, and changes in fat composition [46, 47]. The variations in the effectiveness of vitamin D3 supplementation in terms of its protective role in the hippocampus versus the rest of the body seem to arise from the multifaceted impacts of DEXA on metabolism. These impacts primarily focus on reducing fat mass, inducing muscle wasting, causing intramuscular fat deposition, and leading to mitochondrial dysfunction [48, 49]. The mechanism(s) of DEXA's action resemble stress responses and are associated with hormone dysregulation, insulin resistance, hepatic hexokinase inhibition, and inhibition of glucose oxidation. The consequence of subsequent exposure to GCs leads to increased lipolysis and the release of free fatty acids (FFAs) from adipose tissue, elevated proteolysis of skeletal muscle (decrease in PI3K markers, PKB/Akt, GSK3) dependent on the activation/inhibition of AMP-activated protein kinase (AMPK) by GCs. Reported evidence after 6 weeks of vitamin D3 supplementation in patients with hypercortisolemia had an effect on lipid profile and insulin sensitivity but did not significantly affect baseline body weight (BMI, waist circumference) [50].

Guarnotta V, Di Gaudio F, Giordano C. Vitamin D Deficiency in Cushing's Disease: Before and After Its Supplementation. Nutrients. 2022 Feb 25;14(5):973. doi: 10.3390/nu14050973. PMID: 35267948; PMCID: PMC8912655.

Ferraù F, Korbonits M. Metabolic comorbidities in Cushing's syndrome. Eur J Endocrinol. 2015 Oct;173(4):M133-57. doi: 10.1530/EJE-15-0354. Epub 2015 Jun 9. PMID: 26060052.

Alev K, Aru M, Vain A, Pehme A, Kaasik P, Seene T. Short-time recovery skeletal muscle from dexamethasone-induced atrophy and weakness in old female rats. Clin Biomech (Bristol, Avon). 2022 Dec;100:105808. doi: 10.1016/j.clinbiomech.2022.105808. Epub 2022 Oct 25. PMID: 36368193.

Otsuka Y, Egawa K, Kanzaki N, Izumo T, Rogi T, Shibata H. Quercetin glycosides prevent dexamethasone-induced muscle atrophy in mice. Biochem Biophys Rep. 2019 Feb 11;18:100618. doi: 10.1016/j.bbrep.2019.100618. PMID: 30805562; PMCID: PMC6372881.

Aru M, Alev K, Pehme A, Purge P, Õnnik L, Ellam A, Kaasik P, Seene T. Changes in Body Composition of Old Rats at Different Time Points After Dexamethasone Administration. Curr Aging Sci. 2019;11(4):255-260. doi: 10.2174/1874609812666190114144238. PMID: 30648531; PMCID: PMC6635420.

Koorneef LL, van der Meulen M, Kooijman S, Sánchez-López E, Scheerstra JF, Voorhoeve MC, Ramesh ANN, Rensen PCN, Giera M, Kroon J, Meijer OC. Dexamethasone-associated metabolic effects in male mice are partially caused by depletion of endogenous corticosterone. Front Endocrinol (Lausanne). 2022 Aug 10;13:960279. doi: 10.3389/fendo.2022.960279. PMID: 36034417; PMCID: PMC9399852.

Comment 3. In line 211, Freversed should be reversed.

Response 3. Thank you for indicating this mistake. It has been corrected in our manuscript according to your suggestion.

Line 312-313: “According to our results, systematic vitamin D3 supplementation partially reversed these adverse effects of DEXA.”

Once again, thank you for your time and effort in reviewing our manuscript. We look forward to your final evaluation and hope for the opportunity to share our revised work with the broader scientific community.

Sincerely,

Jan J Kaczor

Round 2

Reviewer 1 Report

Incorporate the findings from the ongoing parallel study into the current research. Ensure that control groups are integrated to enhance the study's validity. It is imperative to include a dedicated vitamin D3 only group for a direct comparison, which is essential for drawing comprehensive conclusions. In such case, you should run western blot analysis including the missed group and repeat the statistical analysis. Isobolograms can help you determine the synergistic, antagonistic or non effects of the combination.

Authors are asked to

1- incorporate the findings from the ongoing parallel study into the current research.

2- Ensure that control groups are integrated to enhance the study's validity. It is imperative to include a dedicated vitamin D3 only group for a direct comparison, which is essential for drawing comprehensive conclusions. In such case, they should run western blot analysis including the missed group and repeat the statistical analysis.

3- Isobolograms can help you determine the synergistic, antagonistic or non effects of the combination.

Author Response

Dear Reviewer,

The authors of the manuscript would like to thank you for all your comments and suggestions concerning improvements of this paper.

Comment

Incorporate the findings from the ongoing parallel study into the current research. Ensure that control groups are integrated to enhance the study's validity. It is imperative to include a dedicated vitamin D3 only group for a direct comparison, which is essential for drawing comprehensive conclusions. In such case, you should run western blot analysis including the missed group and repeat the statistical analysis. Isobolograms can help you determine the synergistic, antagonistic or non effects of the combination.

Response:

We regret to inform you that we are unable to address the suggestions made in the review due to the following reasons:

The experiment described in this publication has been concluded, and we lack the necessary biological material for additional measurements and analysis. This limitation was originally mentioned in the first version of the manuscript under "Limitations of the Study."

The results from our parallel experiment are integral to another paper, making it impossible to include them as part of this article. However, if you request it, we can incorporate data related to body weight and hippocampus mass, as demonstrated in our "first response" to you.

We acknowledge the reviewer's valid point regarding the absence of a control group receiving vitamin D3 to assess the additive effects of simultaneous substance administration. This indeed represents a significant limitation in our study. We can add this information to the manuscript.

Although we believe that our results make a valuable contribution to the current state of knowledge allowing the experimental set-up to be expanded in further studies by adding a control group receiving vitamin D3.

Sincerely,

Jan J Kaczor

Reviewer 3 Report

The authors provided an explanation for the lack of control in Western blot (Figures 4 and 5 ). However, the lack of control puts a doubt on the statistics.  The BCA protein kit can give an accurate total protein concentration but cannot give a relative quality for each protein marker. 

Author Response

Dear Reviewer,

The authors of the manuscript would like to thank you for all your comments and suggestions concerning improvements of this paper.

Comment 1. The authors provided an explanation for the lack of control in Western blot (Figures 4 and 5). However, the lack of control puts a doubt on the statistics. The BCA protein kit can give an accurate total protein concentration but cannot give a relative quality for each protein marker.

Response 1. Thank you for your feedback. The Stain-Free imaging method can eliminate the use of the control proteins. Recent studies suggest that Stain-Free imaging is an efficient, precise method to normalize proteins from a specific sample. The technique has been shown to match methods for normalizing the counter proteins actin and beta-tubulin [Malow et al. 2022]. The most significant advantage of using Stain-Free imaging is the increased precision of normalization, better reproducibility of results, and decreasing sample size of up to 80% compared to normalization to control proteins.

Maloy A, Alexander S, Andreas A, Nyunoya T, Chandra D. Stain-Free total-protein normalization enhances the reproducibility of Western blot data. Anal Biochem. 2022 Oct 1;654:114840. doi: 10.1016/j.ab.2022.114840. Epub 2022 Aug 2. PMID: 35931182; PMCID: PMC10214384.

Stain-Free imaging eliminates the conventional and troublesome practice of relying on housekeeping proteins as loading controls in western blots. Instead, it empowers users to achieve genuine quantitative western blot data by normalizing the bands to the total protein content in each lane.

Sincerely,

Jan J Kaczor
